# Seasonal recurrence and modular assembly of an Arctic pelagic marine microbiome

**Taylor Priest** [1,2,8] ✉, **Ellen Oldenburg** [3,8], **Ovidiu Popa**[3], **Bledina Dede** [1,4], **Katja Metfies** [5], **Wilken-Jon von Appen** [5], **Sinhué Torres-Valdés** [5], **Christina Bienhold** [1,5], **Bernhard M. Fuchs** [1], **Rudolf Amann** [1], **Antje Boetius** [1,5,6] & **Matthias Wietz** [1,5,7] ✉

Deciphering how microbial communities are shaped by environmental variability is fundamental for understanding the structure and function of ocean ecosystems. While seasonal environmental gradients have been shown to structure the taxonomic dynamics of microbiomes over time, little is known about their impact on functional dynamics and the coupling between taxonomy and function. Here, we demonstrate annually recurrent, seasonal structuring of taxonomic and functional dynamics in a pelagic Arctic Ocean microbiome by combining autonomous samplers and in situ sensors with long-read metagenomics and SSU ribosomal metabarcoding. Specifically, we identified five temporal microbiome modules whose succession within each annual cycle represents a transition across different ecological states. For instance, *Cand.* Nitrosopumilus, Syndiniales, and the machinery to oxidise ammonia and reduce nitrite are signatures of early polar night, while late summer is characterised by *Amylibacter* and sulfur compound metabolism. Leveraging metatranscriptomes from *Tara Oceans*, we also demonstrate the consistency in functional dynamics across the wider Arctic Ocean during similar temporal periods. Furthermore, the structuring of genetic diversity within functions over time indicates that environmental selection pressure acts heterogeneously on microbiomes across seasons. By integrating taxonomic, functional and environmental information, our study provides fundamental insights into how microbiomes are structured under pronounced seasonal changes in understudied, yet rapidly changing polar marine ecosystems.

Bacteria, archaea and microeukaryotes are the dominant life forms in ocean environments, serving essential trophic roles and mediating the biogeochemical cycling of biologically important elements[1–3]. These microbes drive and respond to changes in their surrounding environment, which results in the assembly of distinct communities over spatial and temporal scales[4,5]. Deciphering how environmental variability shapes population dynamics and the structuring of community dynamics is thus fundamental for

[1]Max Planck Institute for Marine Microbiology, Bremen, Germany. [2]Institute of Microbiology, ETH Zurich, Zurich, Switzerland. [3]Institute for Quantitative and Theoretical Biology, Heinrich Heine University Düsseldorf, Düsseldorf, Germany. [4]Ecologie Systématique Evolution, CNRS, Université Paris-Saclay, Agro-ParisTech, Gif-sur-Yvette, France. [5]Alfred Wegener Institute Helmholtz Centre for Polar and Marine Research, Bremerhaven, Germany. [6]MARUM – Center for Marine Environmental Sciences, University of Bremen, Bremen, Germany. [7]Institute for Chemistry and Biology of the Marine Environment, University of Oldenburg, Oldenburg, Germany. [8]These authors contributed equally: Taylor Priest, Ellen Oldenburg. ✉e-mail: tpriest@ethz.ch; matthias.wietz@awi.de

**Fig. 1 | The WSC mooring site and environmental conditions between July 2016 and July 2020. a** Bathymetric map of the Fram Strait with the mooring location within the West Spitsbergen Current. Arrows represent average current velocities over the four-year sampling period at the average depth of the moored autonomous

sampler (32 m). **b** Water temperature, oxygen saturation, mixed layer depth, chlorophyll concentrations measured from mooring-attached sensors and photosynthetically active radiation (PAR) derived from AQUA-MODIS satellite data. The shaded grey and yellow represent the periods of polar night and day, respectively.

understanding the structure and function of ecosystems and how they respond to change.

Long-term observations have uncovered recurrent and transient dynamics in microbial communities across daily, seasonal and annual timescales. In temperate ecosystems, communities are predominantly structured by seasonal variability, with broadly recurrent fluctuations of taxa on annual scales[6–11]. These patterns have led to the conclusion that microbial responses to biological and environmental shifts are predictable[12]. However, recent evidence indicates that micro-eukaryotes exhibit weaker temporal structuring than prokaryotes[13], suggesting different controlling mechanisms. Furthermore, high-frequency sampling has shown that population dynamics are highly ephemeral, undergoing rapid, short-lived fluctuations that transpire over days[14–16]. Thus, while microbial communities are structured over time, they also undergo constant flux, reflecting the dynamic nature of ocean ecosystems.

Despite the wealth of knowledge gained from long-term observations, the focus primarily on taxonomic dynamics and on temperate and subtropical ecosystems has left many questions unanswered. In particular, it remains unclear how compositional shifts across environmental gradients translate to changes in the functionality of microbial communities. Since distantly related organisms can perform similar metabolic functions[17,18], taxonomic information alone is a relatively weak proxy for ecological landscapes or ecosystem function. Therefore, long-term observations that integrate taxonomic, functional, and environmental information are greatly needed. This is particularly important in the polar oceans, where long-term observations are rare and unprecedented changes are taking place because of climate warming.

To address this, we investigated the dynamics of prokaryotic (here, this term is used operationally to refer to Bacteria and Archaea) and microeukaryotic communities from a taxonomic and functional perspective over a four-year period in an Arctic Ocean ecosystem, the West Spitsbergen Current (WSC). The WSC constitutes the primary entry route for Atlantic water into the Arctic Ocean and is characterised by pronounced seasonal variability in environmental conditions, archetypal of high-latitude ecosystems. However, unlike other Arctic Ocean ecosystems, the WSC remains ice-free year-round, due to the warm North Atlantic water. Therefore, the WSC represents a model system for investigating the dynamics of microorganisms and ecosystem function in the context of pronounced seasonal variability. In

addition, it can provide valuable insights into potential future shifts in Arctic Ocean ecosystems, given the progressive northward expansion of Atlantic water influence[19]. By leveraging moorings fitted with autonomous sampling and measuring devices combined with 16S and 18S rRNA gene amplicon and PacBio HiFi metagenome sequencing, we continuously tracked taxonomic, functional and environmental dynamics over time. Using a Fourier transformation-based approach, we demonstrate that prokaryotic and microeukaryotic communities exhibit annually recurrent, seasonally structured dynamics that are characterised by a succession across five distinct temporal modules. Each module features unique taxonomic and metabolic signatures, representing distinct ecological states, and are subject to different environmental selection pressures. Our study provides a multi-year ecosystem catalogue from the Arctic that integrates taxonomic, functional and environmental information, and provides fundamental insights into the dynamics and structuring of microbiomes across pronounced environmental gradients.

## Results and discussion
### The West Spitsbergen Current shows pronounced temporal structuring

We first investigated how environmental conditions in the WSC are structured over intra- and inter-annual scales. For this, we combined data collected from in situ sensors, including temperature, salinity, oxygen saturation and chlorophyll $a$, with satellite-derived values of photosynthetically active radiation (PAR) (Supplementary Data 1). Our measurements were derived from the epipelagic layer but, due to the movement of the mooring by currents, sampling depths varied from 20 to 100 m. Owing to the inclusion of CTD sensors at different depths, we were also able to determine the lower bound of the mixed layer depth (MLD). Each annual cycle was characterised by pronounced shifts in environmental conditions (Fig. 1). As expected, these shifts followed the transition between polar night and polar day, which is a major force stimulating biological dynamics in the Arctic[20]. The end of the polar night was marked by an increase in PAR in April, which continued to rise until a maximum of, on average, 38 μmol photons m$^{-2}$ s$^{-1}$ in June. The increasing PAR also coincided with rising water temperatures, which peaked at ~7 °C in August/September before decreasing again to <4 °C between December–May. Changes in MLD were inversely related to temperature (Pearson correlation: $R = -0.47$, $p < 0.05$), with abrupt events of shallowing to <5 m in June and

deepening to >200 m between December and January. Chlorophyll concentrations showed a lagged association with PAR, peaking between June and August (Fig. 1b). However, the magnitude and timing of the chlorophyll peak varied across years, from 3.35 µg L$^{-1}$ in July 2019 to 13.22 µg L$^{-1}$ in June 2018, indicating differences in phytoplankton bloom phenologies. The WSC thus exhibits pronounced intra-annual shifts in environmental conditions, presenting an ideal ecosystem to study seasonally driven biological dynamics.

## The intra- and inter-annual temporal structuring of communities

We next investigated the temporal structuring of prokaryotic and microeukaryotic communities from a taxonomic perspective. Using autonomous Remote Access Samplers (RAS), we collected 97 samples at, on average, fortnightly resolution that were used for 16S and 18S rRNA gene sequencing (Supplementary Data 2). From these, we recovered 3629 bacterial, 119 archaeal and 3019 microeukaryotic amplicon sequence variants (ASVs) (Supplementary Data 3, 4).

The alpha diversity of prokaryotic and microeukaryotic communities exhibited distinct trends within each annual cycle (Figs. 2a, b and Supplementary Data 5). For prokaryotic communities, we observed a tight coupling between Species Richness ($R$), Evenness ($E$) and Shannon Diversity ($H'$), evidenced through significant positive Pearson's correlations (Fig. 2c and Supplementary Data 6). The alpha diversity metrics followed a unimodal fluctuation within each annual cycle for prokaryotic communities, reaching a peak during polar night (December−March) with a mean $R$ of $1210 \pm 208$, $E$ of $0.78 \pm 0.04$ and $H'$ of $5.3 \pm 0.36$. The higher diversity in winter mirrors previous observations in temperate and polar regions[21–23] and is associated with the deepening of MLD, which drives the mixing and dilution of previously stratified communities[24]. The influx of PAR and rapid shallowing of MLD at the end of polar night coincided with a sharp drop in alpha diversity, with the lowest values observed in June ($R = 762 \pm 156$, $H' = 4.50 \pm 0.34$, $E = 0.68 \pm 0.03$). Hence, shifts in prokaryotic alpha diversity were tightly coupled to MLD, supported by strong positive Pearson's correlations; $R$ ($r = 0.71$, 95% CI [0.59,0.79], $p = 4 \times 10^{-16}$), $E$ ($r = 0.43$, 95% CI [0.25,0.58], $p = 1.08 \times 10^{-5}$) and $H'$ ($r = 0.58$, 95% CI [0.43,0.70], $p = 5.7 \times 10^{-10}$) (Supplementary Data 6). This observation is in contrast to previous reports of temperature[25–27], ocean currents[28] and day length[10,22] as the primary drivers of epipelagic bacterial diversity. The greater role of stratification in shaping prokaryotic diversity observed here may be a feature unique to polar oceans, where seasonal shifts in MLD are more pronounced[29] and can be influenced by sea-ice dynamics[30]. However, MLD has rarely been measured or incorporated before, and thus its consideration in future studies will help to ascertain whether its influence varies across latitudes.

In contrast to prokaryotes, microeukaryotic communities exhibited a bimodal fluctuation in alpha diversity in each annual cycle. The bimodal pattern was reflected in a peak in $H'$ in both polar night (February-March) and polar day (July-August). However, during polar night, the increased $H'$ was underpinned by a reduced $R$ and increased $E$, compared to an increased $R$ during polar day. These temporal fluctuations in microeukaryotic diversity are in contrast to observations from the temperate San Pedro time-series (SPOT), where $H'$ was shown to be invariable over time across the top 500 m of the water column[13]. The difference may indicate a stronger seasonal structuring of microeukaryotic communities in high-latitude ocean ecosystems as a consequence of pronounced environmental variability, particularly sunlight and mixing, supported by the negative correlation of MLD and $R$ of microeukaryotes (Pearson's $R = -0.48$, 95% CI [−0.62,−0.31], $p = 7.2 \times 10^{-7}$). Furthermore, the distinct patterns observed in prokaryotes and microeukaryotes provide evidence that the diversity of these communities is shaped by different forces.

We next assessed how the composition of communities was structured over time. We observed a coherent structuring of prokaryotic and microeukaryotic communities based on the month of sampling. That is, communities sampled from the same month across years were typically more similar than from other months in the same year (Fig. 3a, b), reflecting an annual ecosystem clock. For prokaryotic communities, this pattern aligns with previous observations, including month-based clustering at the Banyuls Bay Microbial Observatory in the Mediterranean[31] and lowest pairwise beta diversity values at 12-month intervals reported from the temperate English Channel L4[32] and SPOT[13] time-series. However, in contrast, microeukaryotic communities showed weaker temporal structuring over multi-annual scales at SPOT[13]. By comparing the within- and between-month dissimilarities across years through the convex hull areas within the NMDS ordination (Fig. 3 and Supplementary Data 7), we demonstrate a clear distinction in the temporal recurrence of prokaryotic and microeukaryotic composition. Prokaryotic communities were more cohesive across years in February-March and more variable during June-July, with convex hull areas of ~0.025 and ~0.13, respectively. In contrast, microeukaryotic communities were more cohesive during August and more variable during January-March, with convex hull areas of 0.06 and ~0.43, respectively. However, maximal inter-annual differences in microeukaryotic communities were observed in April, in the phase before the spring bloom. This indicates that the assembly of the spring bloom is not predictable, and only later in the summer, the increased richness of microeukaryotic populations (high $R$) assembles into a cohesive, annually recurrent structure. In previous years, a high inter-annual variability during polar day has been observed in microeukaryotic communities in this region[33,34], so this pattern may change with time and reflect climate-induced or natural decadal variations[34]. The recurrent structuring of microeukaryotic communities during summer could be anticipated to stimulate predictable dynamics in prokaryotes, owing to specialised substrate niches and specific interactions[35,36] as observed in temperate systems[7,8]. Indeed, it triggers a pronounced response of a minority of prokaryotic populations, evidenced through reduced $R$ and $E$, but their emergent population structure shows high inter-annual variability, which may be a result of selection on a functional level and stochastic processes. In contrast, polar night manifests species-rich and compositionally cohesive prokaryotic communities, suggesting that vertical mixing drives the replenishment of the microbiome back to a "standing stock".

## Temporal dynamics of ASVs and community gene content

To gain a deeper understanding of ecosystem dynamics, we complemented the amplicon dataset with long-read metagenomes and investigated temporal fluctuations of ASVs and community gene content. We generated PacBio HiFi metagenomes from 47 time points spanning the 4 years (Supplementary Data 2), resulting in 8.7 million HiFi reads with an average length of 6 kbp. Domain-level classification of the HiFi reads revealed that the majority, 7.9 million, were associated with Bacteria, while 0.47 million were associated with Archaea and 0.27 million with Eukarya. Owing to the length and quality of HiFi reads and the loss of information that typically occurs during metagenome assembly, we elucidated the genetic content of the WSC microbiome by extracting genes directly from the reads. Through this, we recovered 48 million gene sequences that were grouped into 306,088 non-singleton clusters based on a 95% average nucleotide identity cut-off. Combining the gene cluster collection with the ASV data provides a rich multi-year ecosystem catalogue to elucidate community assembly processes and temporal dynamics on a taxonomic and gene content level.

To unravel the temporal dynamics of ASVs and gene clusters, we employed an approach based on Fourier transformations and the determination of oscillation signals. Fourier transformations convert

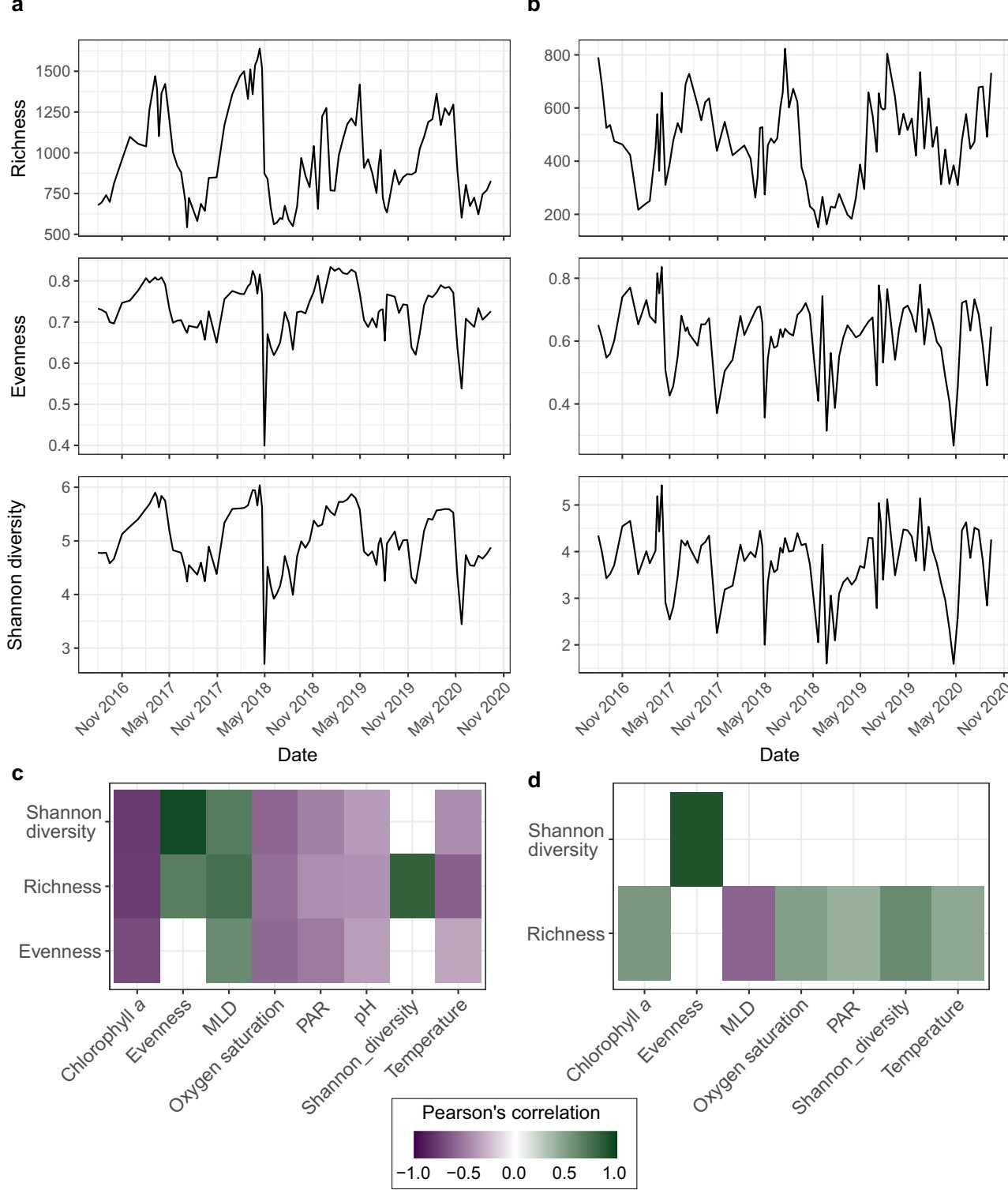

**Fig. 2 | The diversity of prokaryotic and microeukaryotic communities is structured differently over time.** All values of diversity shown represent the mean value after performing 100 iterations of rarefying and metric calculation. Richness, Evenness and Shannon diversity of **a** prokaryotic and **b** microeukaryotic communities. Associations between diversity indices and environmental conditions were tested through Pearson's correlations. Statistically significant associations ($p < 0.05$) after applying Benjamini–Hochberg multiple testing correction are visualised for **c** prokaryotic and **d** microeukaryotic alpha diversity indices.

abundance data into frequencies, resulting in wave-like signals that can be evaluated in terms of peak/trough dynamics, hereon termed oscillation signals. We determined that 18% of prokaryotic ASVs, 15% of microeukaryotic ASVs and 58% of gene clusters exhibited a single oscillation each year, reflecting a unimodal fluctuation in abundance with a single peak and trough (Fig. 3c). Although only capturing a

fraction of the diversity, these annually oscillating ASVs and gene clusters comprised a large proportion of communities across all time points, with an average relative abundance of 67% for prokaryotic ASVs, 55% for microeukaryotic ASVs, and 28% for gene clusters. These findings expand on previous reports of seasonally recurrent patterns in temperate prokaryotic communities at more coarse-grained resolutions,

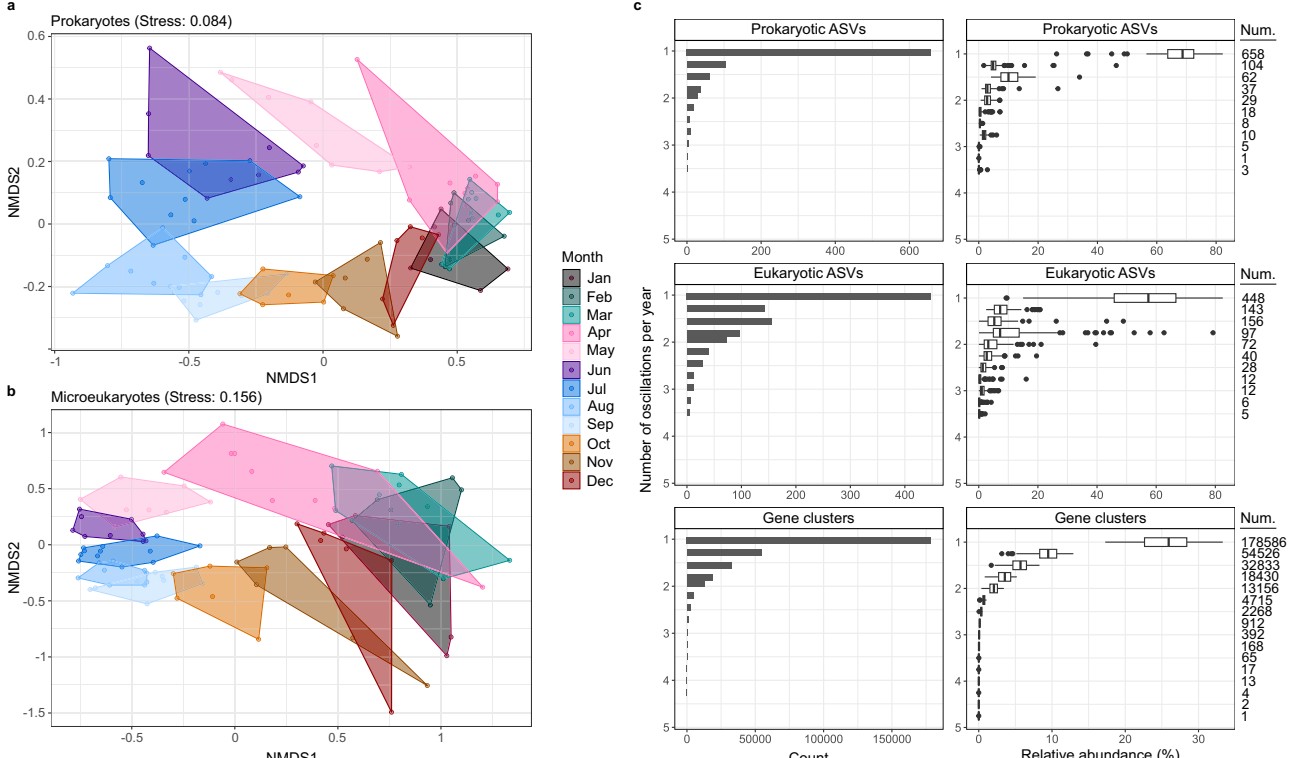

**Fig. 3 | Structuring of community composition and the oscillations of ASVs and genes across years.** Non-metric multi-dimensional scaling of Bray-Curtis dissimilarities calculated after rarefying and Hellinger transforming ASV count data for **a** prokaryotic and **b** microeukaryotic communities over 4 years, wherein colours indicate months. **c** The count (left) and relative abundance (right) of prokaryotic ASVs, microeukaryotic ASVs and gene clusters for each oscillation signal. The relative abundances of ASVs and genes were subject to Fourier transformation, and oscillation signals were determined based on the amplitude of peak/trough dynamics within each annual cycle. An oscillation of 1 indicates a single peak/trough per year, reflecting a unimodal annual fluctuation in abundance. Boxplots illustrate the median, 25 and 75% percentile of the number and relative abundance of ASVs and gene clusters exhibiting each oscillation signal. The number of ASVs and gene clusters observed for each oscillation signal are included next to each boxplot.

such as OTUs[10] and higher taxonomic levels[13], and are in line with those on microdiversity temporal dynamics from a coastal temperate region[8]—suggesting comparable ecological forcing in polar oceans.

The dominance of annually oscillating ASVs raises the question, "To what extent are these patterns deterministic?". More than 15 years ago, Fuhrman and colleagues proposed that "annually recurrent microbial communities can be predicted from ocean conditions"[12]. However, only more recently, with technological advancements and the growing quantity and resolution of data, is predictive ecology becoming more feasible. To contribute to this, we assessed the timing and order of ASV oscillations across each annual cycle. We found that 20% of prokaryotic ASVs consistently oscillated in the same order each year, while 51% reached their peak within the same 30-day window. Therefore, while recurrent dynamics of ASVs are largely bound within temporal windows, the composition of co-occurring populations can vary across years, likely reflecting the influence of trophic interactions and the ephemeral nature of population dynamics[14,37]. A similar pattern was recently described from prokaryotic communities in the NW Mediterranean, where seasonally recurrent taxa exhibited changes in their connections across years[31]. Despite these variations, the periodic timing of recurrent population dynamics within narrow temporal windows provides support towards deterministic patterns in ocean ecosystems—at least in regions that experience marked environmental variability over seasonal timescales, e.g. polar and temperate regions.

## Seasonal recurrence is underpinned by transitions across distinct ecological states

Upon identifying annually oscillating ASVs and gene clusters, we next sought to elucidate how these patterns translate to ecological shifts within the WSC microbiome. To achieve this, integrated taxonomic and functional information is essential. Although taxonomy and function are typically connected through metagenome-assembled genomes (MAGs) in metagenomic studies, this often only captures a fraction of the sequenced material and thus can limit our perspective on microbiome dynamics. This is particularly true in this study, with the recovered 91 species-representative MAGs only capturing between 2.5–35.2% of sequenced genomes (Supplementary Fig. 1 and Supplementary Data 8), largely due to low sequencing depth. An alternative strategy, which is enabled by long-read sequencing, is to directly assess taxonomy and function on a read level. To this end, we performed taxonomic classification on PacBio HiFi reads using a blast and last-common ancestor approach with the GTDB as a reference, and leveraged this information to assign taxonomic labels to gene clusters. Through this, 88% of annually oscillating gene clusters were assigned a phylum-level taxonomy, 84% were assigned a class and 56% were assigned a genus. To translate gene clusters to functional dynamics, we annotated the annually oscillating gene clusters against the EGGNOG database. Of the 178,586 annually oscillating gene clusters, 68% were assigned to reference orthologs and were subsequently grouped into clusters based on the functional annotation of the matching references. This resulted in 7629 unique functional clusters whose oscillations were determined over time.

To investigate temporal structuring on a taxonomic and functional level, we combined the oscillations of functional clusters and ASVs to build a correlation-based network. Correlation co-occurrence networks are powerful for disentangling community dynamics and organismal interactions over spatial and temporal scales[38]. Here, we compared the oscillation signals of ASVs and gene functions through

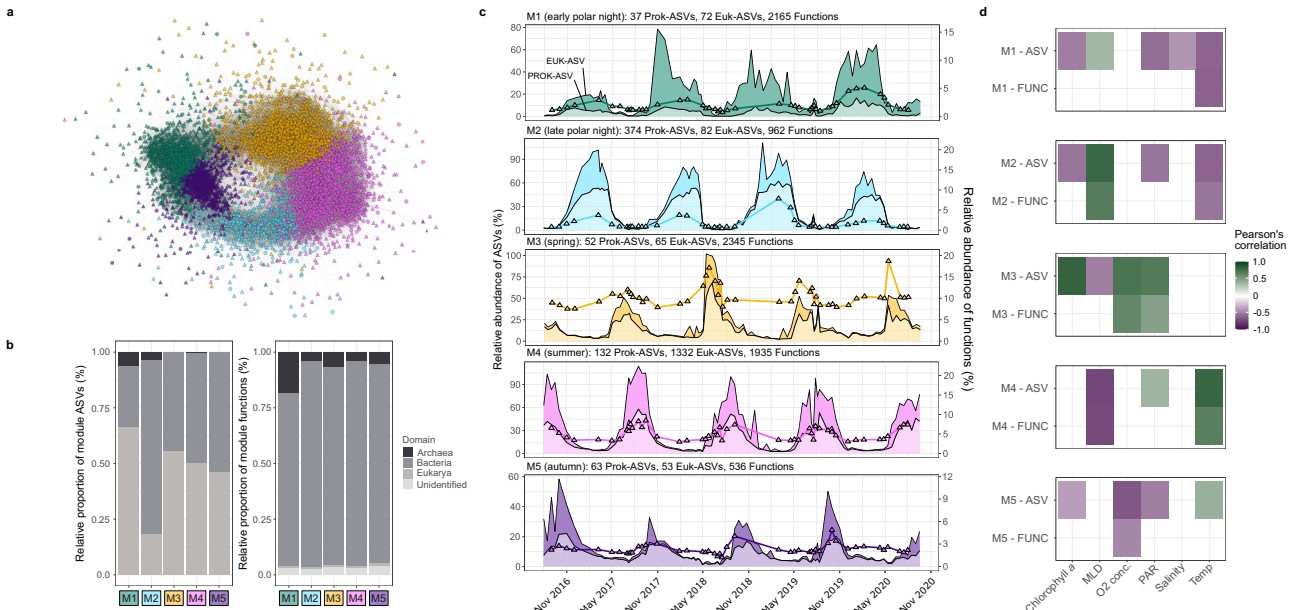

**Fig. 4 | Ecosystem modules, their temporal dynamics and association with environmental conditions. a** Co-occurrence network constructed from significant positive Pearson correlations (>0.7) between prokaryotic, microeukaryotic ASVs and functional cluster oscillation signals. Circle nodes = ASVs, Triangle nodes = functional clusters. **b** The relative proportion of module ASVs and functions associated with the three domains of life. **c** The temporal dynamics of module prokaryotic ASVs (lighter-coloured area), microeukaryotic ASVs (darker-coloured area) and functional clusters (line). **d** Pearson's correlations between the combined relative abundance of module ASVs and module functional clusters against measured environmental factors (only those with a $p < 0.05$ after Benjamini–Hochberg multiple testing correction are shown).

Pearson's correlation and retained only those with a strong, positive coefficient ($R > 0.7$, FDR-based adjusted $p < 0.05$). We subsequently built a network using these correlation coefficients as the edges and the respective ASVs and functions as the nodes. Using the Louvain algorithm, we partitioned the network into five modules comprised of co-oscillating ASVs and functions (Fig. 4a) that represent distinct temporal states in the WSC. Thus, each annual cycle was characterised by a succession across these temporal modules (Fig. 4c).

To uncover the ecological shifts associated with module succession, we compared the composition of modules and the environmental conditions under which they prevail. The modules differed markedly in their diversity and composition of taxonomic and functional components (Fig. 4b), from a low number of ASVs (prokaryotic = 37, eukaryotic = 72) and high number of functions ($n = 2165$) in the early polar night module M1 to a high number of both ASVs (prokaryotic = 132, eukaryotic = 1332) and functions ($n = 1935$) in the summer module M3. Exploring these patterns further revealed clear differences in the distribution of ASVs and functions attributed to *Archaea*, *Bacteria* and *Eukarya* across the modules. The functional composition of modules was dominated by *Bacteria* year-round, although a threefold enrichment of archaeal functions (18% of module functions) was observed during early polar night (module M1). From a taxonomic perspective, eukaryotic ASVs dominated the early polar night (M1), spring (M3) and summer modules (M4), while recurrent bacterial ASVs were more prominent during late polar night (M2) and autumn (M5). To ensure that these patterns were reflective of biological signals and not a consequence of uneven sequencing depth across metagenomes or the frequency of sampling across seasonal periods, we assessed whether the composition of modules changed after subsampling to only one sample per month and rarefying the ASV and functional cluster datasets. The subsampling had a negligible impact on the diversity and composition of modules (Supplementary Fig. 2). Although this does not exclude that additional sampling may further resolve these patterns, it supports the taxonomic and functional differences observed between modules.

The temporal succession of modules represents a transition across distinct ecological states within each annual cycle. These ecological shifts are evidenced through the unique taxonomic (Fig. 5a) and metabolic signatures of each module (Fig. 5b). The early polar night module M1 was taxonomically distinguished by *Cand*. Nitrosopumilus, SAR86, *Porticoccus* and *Syndiniales* and functionally distinguished by nitrogen metabolism, specifically ammonia oxidation (*amoABC*) and nitrite reduction (*nirK* and *norBC*), as well as carbon fixation machinery of the hydroxypropionate/hydroxybutyrate cycle[39]. Read taxonomic classification supported that the ammonia oxidation and carbon fixation machinery was attributed to members of *Nitrosopumilaceae*. The dominance of *Cand*. Nitrosopumilus ASVs during this period and their large degree of connectedness in the network to other ASVs and functions within module M1 (Supplementary Fig. 3) suggests their integral role in polar night microbiomes. This finding expands on previous observations of higher *Cand*. Nitrosopumilus abundances during polar night in southward flowing Arctic waters[40] and increased ammonia oxidation rates in Antarctic coastal waters[41]. A shift in dominance from module M1 to M2 marked an ecological transition during mid- to late-polar night, underpinned by the emergence of *Thalassobius*, *Arenicellaceae* and Radiolarians group C. Despite this taxonomic shift, few distinguishing functional features could be elucidated − resolving the functional transition from early to late-polar night may require increased sampling resolution and the incorporation of additional strategies to improve functional annotations of genes, such as protein structure-based comparisons. The onset of solar radiation and rapid transition from module M2 to M3 marks the end of the polar night. Module M3 represents the early spring phytoplankton bloom, supported by correlations with PAR and chlorophyll (Fig. 4c and Supplementary Data 9) combined with the emergence of photosynthetic organisms (*Pseudo-nitzschia* and *Micromonas*). Thus, this module was dominated both by phototrophs and heterotrophic bacteria, with the latter recognised as primary responders to phytoplankton blooms and their carbohydrate exudates in temperate regions[7,35], including *Polaribacter*, *Aurantivirga* and SAR92.

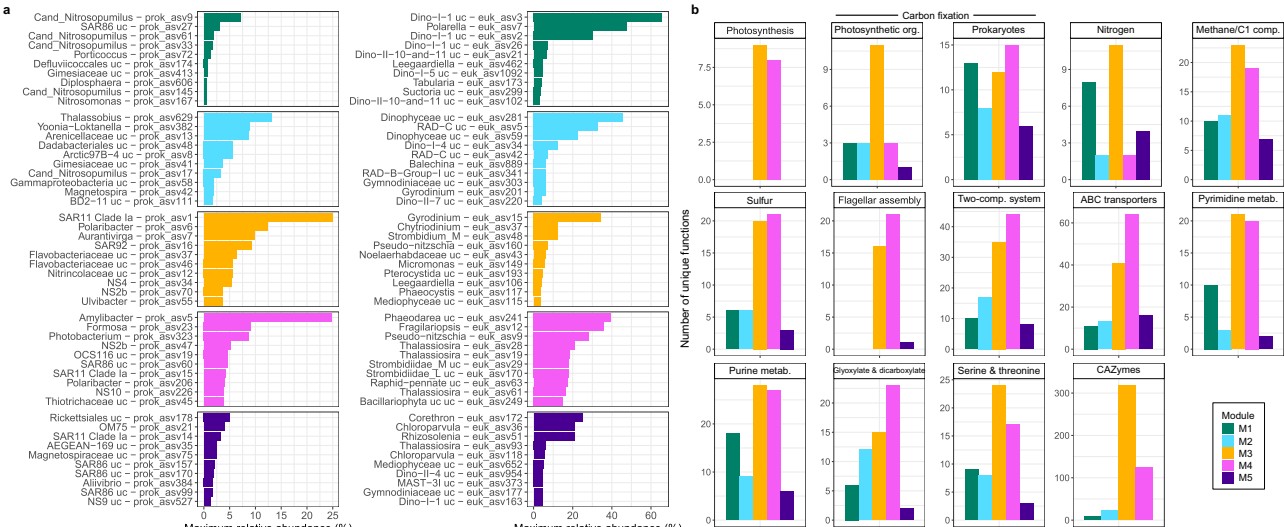

**Fig. 5 | Modules are phylogenetically and functionally distinct. a** The maximum relative abundance of the ten most abundant prokaryotic and microeukaryotic ASVs in each module. **b** Composition of KEGG metabolic pathways that exhibited the largest variance in the number of assigned functions between modules along with the number of carbohydrate-active enzyme (CAZyme) gene families.

Functionally, module M3 was enriched in photosynthesis machinery, nitrogen metabolism (glutamate synthesis and degradation and the *nrtABC* nitrate/nitrite uptake system), carbohydrate-active enzymes and organosulfur compound utilisation, including DMSP demethylation (*dmdBCD*), DMSO to DMS conversion (*dmsBC*), taurine uptake (*tauABC*) and sulfite reduction (*soeABC*). Taurine uptake machinery was almost exclusively attributed to *Alphaproteobacteria*, particularly *Pelagibacteraceae* and *Rhodobacteraceae*, while the machinery to convert DMSO to DMS was attributed to *Flavobacteriaceae* and *Opitutales* and the sulfite reduction machinery to *Rhodobacteraceae* and *Pseudomonadales*.

Sulfur metabolism persisted as a key feature in the summer module M4 but transitioned to both inorganic and organic sulfur utilisation. In particular, summer was characterised by the machinery for sulfur oxidation (*soxABCXYZ*), DMSP demethylation (*dmdA*), methanethiol oxidation (*MeSH*), and alkane sulfonate utilisation (*msuD*), which was accompanied by an enrichment in flagellar assembly, two-component system machinery and ABC transporters. The prokaryotic community in module M4 was dominated by *Amylibacter*, along with other heterotrophic bacteria affiliated with *Formosa* and NS2b, while the eukaryotic community was comprised of diverse functional groups, including *Fragilariopsis* (photosynthetic diatoms) and *Phaeodarea* (heterotrophic protists). *Amylibacter* are key contributors to organosulfur compound metabolism in temperate coastal waters during summer[42,43]. However, we could also attribute the machinery to oxidise sulfur (SOX system), demethylate DMSP (*dmdA*) and reduce sulfite (*soeAB*) to *Amylibacter*, based on read taxonomic classification against GTDB representatives, suggesting their capacity to utilise diverse sulfur sources[44].

The taxonomic and metabolic signatures of each module demonstrate that the WSC microbiome is structured by a succession across five distinct ecological states within each annual cycle that is largely coupled to changes in environmental conditions. Previous taxonomic-centred analyses of prokaryotic communities from temperate and tropical ecosystems have also reported recurrent dynamics structured by seasonality. However, the proportion of ASVs exhibiting seasonal recurrence and the number of temporal modules appears to be higher in the WSC. For instance, in the NW Mediterranean, only 4% of prokaryotic ASVs, constituting a relative abundance of 47%, showed seasonal recurrence and could be grouped into three distinct seasonal clusters[45]. In a coastal temperate region that lacks pronounced

phytoplankton blooms but experiences large environmental variability, recurrent dynamics of prokaryotes primarily partitioned into summer and winter groups[46]. The higher prevalence of recurrent dynamics and their organisation into narrower temporal modules in the WSC may indicate stronger selective pressure arising from the more pronounced seasonal environmental variability.

## Functional dynamics in the WSC are reflective of microbiomes across the wider Arctic Ocean

We next assessed whether the ecological states in the WSC are representative of microbiomes across the wider Arctic Ocean. After passing through the Fram Strait, the WSC turns east and integrates into the Arctic circumpolar boundary current, which circulates along the continental slope at intermediate water depths (>200 m)[47]. The Atlantic water within the boundary current later separates from the slope due to pronounced bathymetric features and subsequently enters the deep central basins. As a result, Atlantic water derived from the WSC has a significant influence across the wider Arctic Ocean—an influence that is currently expanding[19]. Given this, we hypothesised that the WSC can serve as a valuable model system for assessing ecological dynamics within Arctic Ocean microbiomes. To test this, we investigated the activity of module functions across the Arctic Ocean using metatranscriptomes generated from the *Tara Ocean* Polar Circle (TOPC) expedition. We observed that functions within the spring module M3 and autumn module M5 exhibited the highest transcription levels in surface and deep chlorophyll maximum waters (Fig. 6a, b), followed by those of the summer module M4. In contrast, the transcription of functions in mesopelagic depths was dominated by those of the autumn module M5, while signals of spring and summer functions were diminished (Fig. 6b). The only exception to these patterns was the station within the Fram Strait (TARA_163), whose surface waters were dominated by the transcription of functions of the late polar night module M2. The distinct functional composition observed at TARA_163 likely stems from the influence of southward flowing polar water that occurs in this central Fram Strait location (1.4°E), supported by the lower surface water temperatures (<2 °C) compared to the WSC. Exploring the composition of module functions further, we identified that transcription signals of module M3 in surface and deep chlorophyll maximum waters of the Arctic Ocean were dominated by functions associated with photosynthesis and secondary metabolism, in line with previous findings from the same dataset[48], while module

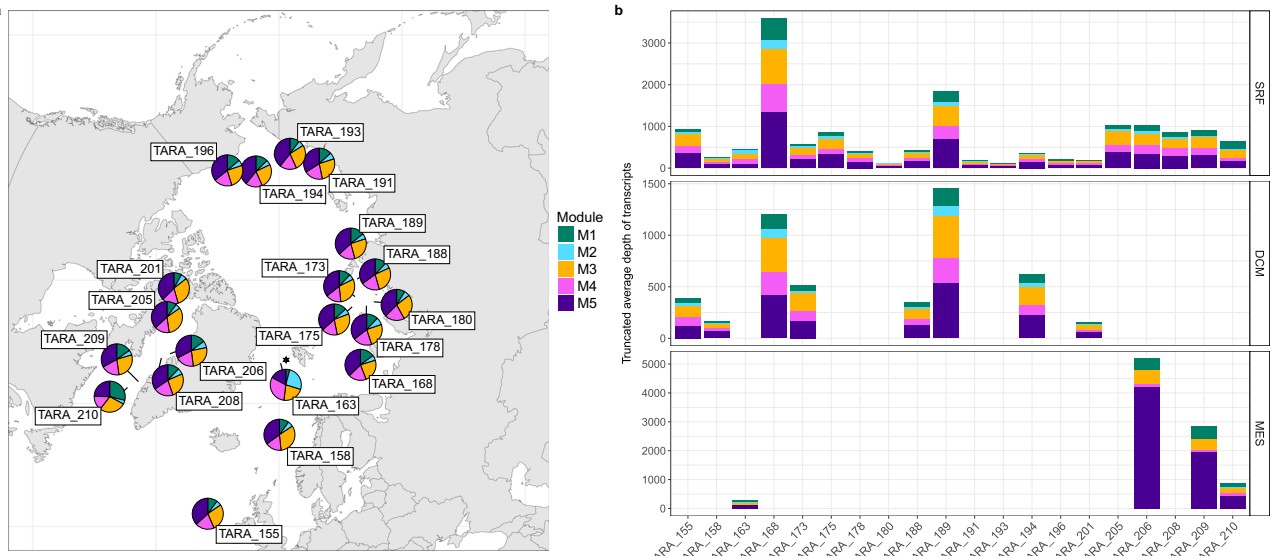

**Fig. 6 | Transcription of module functions across surface, deep-chlorophyll maximum and mesopelagic depth layers of the Arctic Ocean.** Transcription of module functions was determined by read recruitment from metatranscriptomic data generated from surface (SRF), deep-chlorophyll maximum (DCM) and meso-pelagic (MES) depth layers of the Arctic Ocean during the *Tara Ocean* Polar Circle expedition. For each module function, the 80% truncated average depth (TAD80) of transcripts was calculated. The mean of the TAD80 values for all functions in each module was determined, providing a mean of transcription at the module level. **a** Map illustrating the relative proportion of module functions transcription across surface waters at TOPC sampling stations. The black star indicates the location of the RAS mooring. **b** The mean TAD80 of transcripts of module functions across depth layers and stations. Bars not present in DCM and MES layers are where no metatranscriptomic data was available.

M3 functions also dominated the transcription of energy metabolism processes (Supplementary Fig. 5). In contrast, the transcription signals associated with module M5 were primarily underpinned by ribosomal synthesis and transcription/translation machinery. Given that the TOPC expedition took place between May and October, these patterns not only support that the annually oscillating functions in the WSC are actively transcribed across the wider Arctic Ocean but also confirm that they largely proliferate during similar temporal periods. These observations support that the WSC can serve as a valuable system for gaining insights into and inferring the ecological dynamics of Arctic Ocean microbiomes.

## Selection pressure is heterogeneous across seasonal ecological states

The recurrent dynamics of ASVs, genes and functions and their assembly into cohesive, ecological modules suggest that the WSC microbiome is predominantly shaped by deterministic processes. The primary mechanisms that contribute to driving the deterministic structuring of microbiomes are environmental selection and orga-nismal interactions. Environmental selection is considered the primary force shaping the distribution of populations and the assembly of microbial communities in ocean ecosystems[49]. However, we lack a mechanistic understanding of how environmental selection operates. In particular, it is unclear whether the environment primarily selects for a function or for a specific organism with a function. Recent evi-dence has shown that the taxonomic composition of a microbiome can change while the gene content remains conserved[50,51], reflecting the prevalence of metabolic redundancy across microbial taxa. In addition, studies on surface ocean microbial communities have demonstrated incongruencies in structuring on a taxonomic and functional level, with strong selection pressure on functional groups but weak selection pressure on the taxonomic composition within functional groups[18]. These observations provide evidence for the decoupling of taxonomy and function within microbiomes and suggest that selection may act on a functional level.

To assess the influence of selection on shaping microbiome dynamics, we investigated the composition and structuring of gene clusters within functions over time. We focus on the spring module M3 and polar night module M2, as they represent two contrasting sce-narios. We first assessed the proportion of functions comprised of multiple, as opposed to single, gene clusters as a proxy for functional redundancy. In module M3, 85% of functions are comprised of multiple gene clusters, compared to only 67% in module M2, indicating more functional redundancy in spring compared to late polar night (Fig. 7a). Along with increased redundancy, a higher proportion of multi gene-cluster functions exhibited a strong positive linear relationship between function abundance and the diversity of the contained gene clusters in module M3 than module M2, 57 and 32%, respectively (Fig. 7b). That is, functional dynamics in spring are primarily driven by the concurrent growth of metabolically overlapping organisms. Exploring the dynamics of redundant functions further, we observed the same gene cluster dominating each year in 73% of functions during late polar night compared to only 55% in spring (Fig. 7c). Taken toge-ther, these observations suggest that selection pressure is stronger on a functional level in spring.

Integrating the observations of functional dynamics with those of ASVs further supports seasonal variations in environmental selection pressure. The late polar night period is characterised by high species richness and evenness (Fig. 2) with a large number of recurrent populations that assemble into a consistent community structure across years (Fig. 3). Amongst the recurrent populations of late polar night, we also observe taxa that are typically associated with deeper water layers, such as *Dadabacteriales*. Combined with the gene cluster and functional dynamics, these observations suggest that deeper vertical mixing during polar night drives a recurrent homogenisation of epipelagic communities, with either a weak environmental selection pressure or selection acting on an organismal level. In contrast, spring is characterised by low species richness and evenness and high inter-annual variability in beta diversity, driven by only a small assemblage of recurrent populations. The recurrent populations in spring are

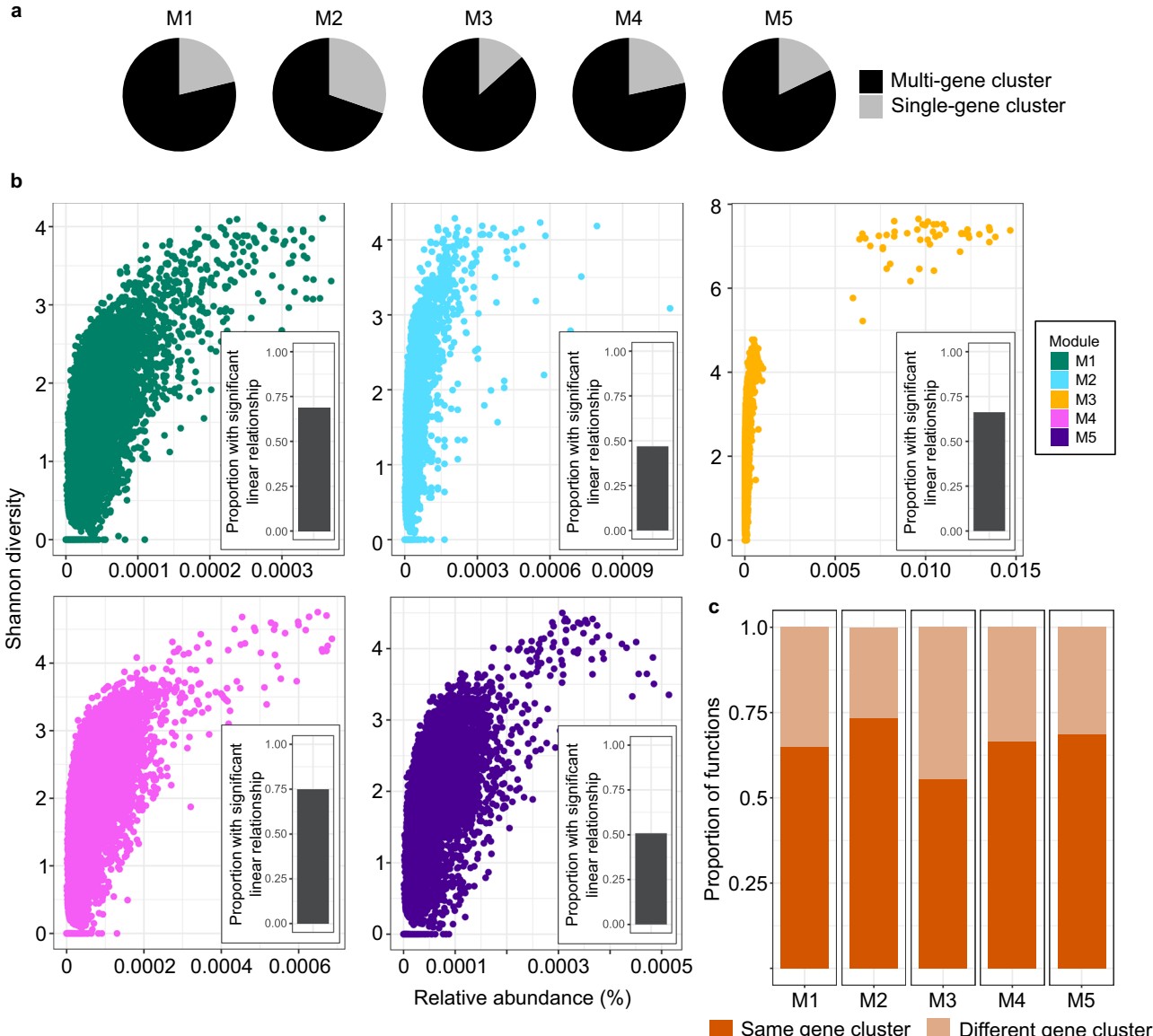

**Fig. 7 | Diversity, abundance and structuring of gene clusters within functions. a** Proportion of functions in each module comprised of multiple or single gene clusters. For those functions comprised of multiple gene clusters, we calculated the **b** Shannon diversity of the gene clusters within each function and compared this to the relative abundance of the function. Each point represents an individual function. The inserted barplots illustrate the proportion of functions comprised of multiple gene clusters that exhibit a significant positive linear relationship between the gene cluster Shannon diversity and function abundance. **c** Proportion of functions that are dominated by the same or different gene cluster when the function reaches peak abundance each year.

functionally redundant, and the same functions are often dominated by different populations each year. Therefore, selection pressure during spring appears to act primarily on the functional level.

To further advance our understanding of environmental selection and the mechanisms that shape the structuring of ocean microbiomes, it is paramount to integrate both taxonomic and functional profiling along with more extensive environmental information. Of particular importance is information on the availability of organic carbon and energy substrates, which play a fundamental role in shaping the dynamics of microbial communities. For instance, phytoplankton blooms in coastal temperate regions that are of a larger magnitude than in the WSC have been shown to drive a predictable composition of microbes[7,8], in contrast to our observations here. The disparity between the regions may be related to the quantity of organic matter released during such phenomena, i.e. higher organic matter concentrations may drive more deterministic taxonomic responses. However, whether the availability of organic matter, or other here

unmeasured environmental factors, plays a role in shaping the deterministic and recurrent dynamics within microbiomes requires further investigation.

Our study provides fundamental insights into the seasonal and inter-annual structuring of prokaryotic and microeukaryotic communities in an Arctic pelagic ocean ecosystem. We demonstrate the prevalence of annually recurrent dynamics of populations and community gene content, which are organised into five distinct seasonal modules. Each of the modules represents a unique ecological state, underpinned by specific prokaryotic and microeukaryotic organisms and functions, and is connected to certain environmental conditions. We further provide evidence that environmental selection is heterogeneous across these ecological states, with differential pressures on an organismal and functional level. Our findings provide fundamental insights into understudied yet rapidly changing ocean regions and advance our understanding of how microbial communities are structured across pronounced environmental gradients.

## Methods

### Sample collection and processing

Moorings carrying autonomous water samplers (Remote Access Samplers; RAS) were deployed between 2016 and 2020 at a single location in the eastern Fram Strait (mooring F4: 79.0118N 6.9648E). Moorings were deployed for 12-month intervals, with collection and redeployment occurring in summer, typically in August. Owing to ocean currents, the vertical positioning of the RAS fluctuated between 20 and 110 m over the 4-year period. At weekly to fortnightly intervals, 2 × 500 ml of seawater was collected in sterile plastic bags and fixed with mercuric chloride (0.01% final concentration). Following mooring recovery, fixed seawater samples from each time point were filtered onto 0.22 μm Sterivex cartridges and directly frozen at −20 °C until DNA extraction.

### Mooring and satellite data

Seabird SBE37-ODO CTD sensors measured temperature, depth, salinity and oxygen, as detailed in a preceding study[23]. Employing multiple CTD sensors along the mooring enabled the determination of the minimum mixed layer depth (MLD) at each sampling time point. For instance, if two CTDs showed the same temperature and salinity measurements, the MLD was at least the depth of the deeper CTD. Chlorophyll concentrations were measured via Wetlab Ecotriplet sensors. Surface water photosynthetically active radiation (PAR) data, with a 4 km grid resolution, was obtained from AQUA-MODIS (Level-3 mapped; SeaWiFS, NASA) and extracted in QGIS v3.14.16 (http://www.qgis.org).

### SSU rRNA gene amplicon and metagenome sequencing

Filtered seawater samples from 97 time points were subjected to DNA extraction using the DNeasy PowerWater Kit (QIAGEN, Hilden, Germany). 16S and 18S rRNA gene fragments were PCR-amplified using the primers 515F–926R[52] and 528iF–964iR[53], respectively. Sequencing libraries were constructed from rRNA gene products according to the "16S Metagenomic Sequencing Library Preparation" protocol (Illumina, San Diego, CA) and sequenced on an Illumina MiSeq platform in 2 × 300 bp, paired-end mode. Amplicon sequencing took place at the Alfred Wegener Institute. The extracted DNA from 47 time points was additionally used to generate PacBio HiFi metagenomes. Sequencing libraries were prepared following the protocol "Procedure & Checklist – Preparing HiFi SMRTbell Libraries from Ultra-Low DNA Input" (PacBio, Menlo Park, CA), followed by inspection with a FEMTOpulse. The libraries were multiplexed and sequenced on 8 M SMRT cells (7–8 samples per cell) on a PacBio Sequel II platform for 30 h with sequencing chemistry 2.0 and binding kit 2.0. Metagenomes were sequenced at the Max Planck Genome Centre, Cologne, Germany.

### Recovery of metagenome-assembled genomes

PacBio HiFi reads from each metagenome were assembled using metaFlye v2.9.1[54] (parameters: --meta --pacbio-hifi --hifi-error 0.01 --keep-haplotypes). The depth of coverage of contigs across all 47 metagenomes was determined through read recruitment using minimap2 v2.1[55] (parameters: -ax map_hifi --sam-hit-only) followed by the jgi_summarize_bam_contig_depths script from MetaBat2 v2.15[56]. Contigs from each metagenome were binned using MetaBat2 with default settings. The completeness and contamination of bins was determined using the lineage_wf workflow of CheckM v1.2.0 and those with a completeness <50% and contamination >10% were removed. The quality-filtered bins from the 47 metagenomes were dereplicated into species-level clusters based on a 95% average nucleotide identity using the dereplicate function of dRep v3.2.2[57] (parameters: -sa 0.95 -nc 0.50 -comp 50 -con 10). The taxonomic classification of species-representative metagenome-assembled genomes (MAGs) was performed against the GTDB r220 database using the classify_wf workflow of gtdbtk v2.3.2[58].

The abundance of species-representative MAGs was determined based on the average depth of coverage of four ribosomal proteins (large subunit 3, large subunit 4, large subunit 6 and small subunit 8). The average depth of the four ribosomal protein genes was used to estimate the number of genomes sequenced in each metagenome. The same sequences, extracted from the raw PacBio HiFi reads, were competitively aligned against the species-representative MAGs using minimap2 v2.1[55] (parameters: -ax map_hifi --sam-hit-only). For each MAG, we calculated the average depth of aligned ribosomal protein gene sequences and took the quotient between this value and the average depth on the metagenomic read level, resulting in values that represent the proportions of genomes sequenced.

### Gene and functional analysis

A total of 48 Gbp of PacBio HiFi reads were generated, with an average of 1 Gbp per sample. Gene sequences were predicted on HiFi reads using Fraggenescan (v1.31; parameters: -complete=1 -train=sanger_5)[59]. Reads and their contained gene sequences were taxonomically classified at the domain-level using Tiara v1.0.3, followed by a diamond blastp and last-common ancestor approach for those identified as *Archaea* and *Bacteria*, as described previously[60]. In brief, a Diamond v2.10[61] blastp search (minimum coverage of 40% and identity of 60%) was performed against a database containing gene sequences predicted from the GTDB species-representatives. The last-common ancestor approach was applied, using the tool taxonkit[62], to provide a conservative taxonomic classification at the gene level and then repeated at the read level. A gene cluster catalogue was constructed through clustering of gene nucleotide sequences at a 95% identity threshold with the *easy-cluster* workflow from MMSeqs2 v15.6[63] (parameters: --min-seq-id 0.95 -c 0.6 --cluster-reassign). To facilitate comparisons between metagenomes, gene cluster counts were normalised by the estimated number of prokaryotic genomes in each sample, determined from the average sequencing depth of 16 single-copy ribosomal proteins, as described previously[60], before converting to relative proportions. The longest sequence from each cluster was used as the representative for a functional assignment against the EGGNOG v5.0 database[64] using the eggnog-mapper tool v2.1.12[65] with a minimum coverage of 60% and identity of 40% (parameters: --query_cover 60 –pident 40 --sensmode more-sensitive --pfam_realign). To build the 'functional clusters' used in our analysis, we grouped gene clusters based on functional annotations of matching seed orthologs. As the seed orthologs of EGGNOG have been functionally annotated using numerous sources, we first grouped the gene clusters that had an annotation from the carbohydrate-active enzyme database, followed by KEGG and then PFAM. The abundance of functional clusters was determined from the sum of the contained gene clusters.

### Taxonomic diversity analyses

The 16S and 18S rRNA gene sequences were processed into Amplicon Sequence Variants (ASV) using DADA2[66] in R v4.1.3[67]. ASVs were taxonomically classified using the SILVA SSU v138 (16S) and PR2 v4.12 (18S) databases using the assignTaxonomy function of DADA2. We only considered ASVs with >3 counts in >3 samples. For alpha diversity, the ASV count table was subject to 100 iterations of rarefying followed by the calculation of Richness, Shannon diversity (*diversity* function in *vegan*) and Evenness. The mean and standard deviation was calculated for each metric, with the mean being used for visualisations and statistical comparisons to environmental metadata. For beta diversity analysis, the ASV count table was rarefied a single time, followed by Hellinger transformation before the calculation of Bray-Curtis dissimilarities (using *vegan*). Dissimilarities were ordinated using non-metric multi-dimensional scaling.

## Time series and network analysis

The temporal analysis of ASVs and gene clusters, as well as the construction of co-occurrence networks, was performed using the same workflow as that presented recently by authors of this paper[68]. In brief, for each ASV and gene cluster, we employed a Fourier transformation approach to calculate a time-series signal from relative abundances using the *segmenTier/segmenTools* packages[69]. The Fourier Transform is a technique for decomposing functions or signals into the sum of their frequency components, characterised by sine and cosine functions. In practice, this involves converting temporal variations in relative abundances into frequencies, whereby each frequency is comprised of two parts, the real components and the imaginary components. The imaginary components are derived from the calculation of the sine and cosine trajectories. Together, these two components describe the amplitude and phase of the oscillation of the frequency. To determine the oscillation signals, i.e. the number of peaks in the frequencies during each annual cycle, we reconstruct the time series through a limited number of significant Fourier components. For this, we retained only the first few Fourier components, which capture the frequencies that contribute the most to the signal. By focusing on these significant components, it minimises noise and minor fluctuations that do not inform about major temporal dynamics. Through this, we obtain the number of oscillations in the frequencies within each annual cycle, which we term oscillation signals. A single oscillation would refer to a single peak in abundance of an ASV or gene cluster within a single annual cycle. The oscillation signals were extracted for each ASV and gene cluster using the *fprocessTimeseries* function from the *segmenTier* package. The ASVs and genes with an oscillation signal of one, i.e. a single peak and trough in each annual cycle, were deemed as annually oscillating and retained for further analysis. Phase-Rectified Signal Averaging (PRSA) was used to visualise periodic patterns of ASVs, using phase-rectified data to remove phase variability. Then, the phase-rectified data was averaged for the final PRSA plot. The oscillation signals of ASVs and gene clusters were compared through pairwise Pearson's correlation, with multiple testing corrections using the FDR method. Those with a statistically significant ($p < 0.05$) positive correlation coefficient of >0.7 were used to build a co-occurrence network, with edges as correlation coefficients. By using oscillation signals, focusing on ASVs and gene clusters with a defined oscillation, and using only positive correlations, we minimise the noise in the dataset and prevent potential network topology distortion of negative correlations. Co-occurrence correlation networks were constructed using the *igraph* package[70] in R and visualised in Cytoscape v3.7.2[71] using the edge-weighted spring-embedded layout. ASVs and gene clusters in the network were clustered using the Louvain algorithm[72]. All steps outlined above were performed in R v4.1.3.

## Statistical analysis of microbiome modules

To compare the diversity of module components while accounting for variations in sequencing depth, we iteratively calculated the diversity of ASVs and gene clusters in each module after subsampling. That is, each iteration involved subsampling the ASV and gene cluster tables to the lowest count and then calculating the diversity of components that were retained in each module. To ensure that the diversity comparisons were not influenced by the frequency of metagenome samples over the time series, we repeated the same subsampling and diversity calculation while only considering one sample per calendar month. The dynamics of ASVs and gene clusters in each module were compared with environmental conditions through Pearson's correlations using the *rcorr* package in R.

## Assessment of module function transcription across the wider Arctic Ocean

To assess the dynamics and activity of module functions across the wider Arctic Ocean, we leveraged a read recruitment approach using metatranscriptomes generated from the *Tara Oceans* Polar Circle expedition. Metatranscriptomic reads were mapped to gene cluster representatives using BBMAP v39.06[73] using a minimum identity threshold of 95%. The coverage of gene clusters in each metatranscriptome was determined from mapped read counts using the 'genomecov' function of the Bedtools v2.31.1 programme[74]. Coverage values were further processed to determine the 80% truncated average depth (TAD80) value for each gene cluster in each metatranscriptome using the BedGraph.tad.rb script from the Enveomics collection[75].

## Reporting summary

Further information on research design is available in the Nature Portfolio Reporting Summary linked to this article.

## Data availability

Mooring data are available at PANGAEA under 904565 (2016–2017), 904534 (2017–2018), 941126 (2018–2019), and 946508 (2019–2020). 16S rRNA amplicon reads are available at ENA under PRJEB43890 (2016–2017), PRJEB43889 (2017–2018), PRJEB67813 (2018–2019), PRJEB66202 (2019–2020). 18S rRNA amplicon reads are available under PRJEB43504 (2016–2017), PRJEB43885 (2017–2018), PRJEB66212 (2018–2019), PRJEB66220 (2019–2020). Raw metagenomic reads, assemblies and metagenome-assembled genomes are available under PRJEB67368. Accessions for each amplicon and metagenome sample are listed in Supplementary Data 2. Source Data necessary to reproduce the analysis and figures presented in this manuscript are available at https://zenodo.org/records/14586634.

## Code availability

The code for reproducing the analysis and recreating the figures presented in this manuscript are available at https://github.com/tpriest0/Fram_Strait_WSC_time_series_2016-2020.

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

## Acknowledgements
We thank Jana Bäger, Theresa Hargesheimer, Rafael Stiens and Lili Hufnagel for RAS operations; Daniel Scholz for RAS and sensor operations; Normen Lochthofen, Janine Ludszuweit, Lennard Frommhold and Jonas Hagemann for mooring operations; Jakob Barz, Swantje Ziemann and Anja Batzke for DNA extraction, library preparation and sequencing; and Bruno Huettel, Christian Woehle and the technicians at the Max Planck Genome Centre in Cologne for sequencing. The captain, crew and scientists of RV Polarstern cruises PS99.2, PS107, PS114, PS121 and PS126 are gratefully acknowledged. This project has received funding from Polarstern grants AWI_PS99_00, AWI_PS107_05, AWI_PS114_01, AWI_PS121_07, AWI_PS126_05, and AWI_PS126_07. Further support came from the European Research Council (ERC) under the European Union's Seventh Framework Programme (FP7/2007-2013) research project ABYSS (Grant Agreement no. 294757) to AB, from the Helmholtz Association, specifically for the FRAM infrastructure, and from the Max Planck Society.

## Author contributions
T.P. designed and executed ASV and metagenomic analyses and wrote the manuscript, with input from M.W., E.O. and B.D. E.O. and O.P. performed time series and network analyses. M.W. processed amplicon raw data into ASVs, co-designed the sampling and mooring strategy, and coordinated data analysis. W.-J.v.A. contributed quality-controlled oceanographic data and coordinated the mooring operations. K.M. coordinated the processing of samples and sequencing and provided 18S rRNA gene sequence data. S.T.-V. provided quality-controlled chlorophyll sensor data. C.B., K.M. and A.B. co-designed the sampling and mooring strategy, and contributed to the interpretation. B.D., B.F. and R.A. contributed to the study design and interpretation. All authors contributed to the final manuscript.

## Funding

## Competing interests
The authors declare no competing interests.
