## [Peer Review file · Nature Communications]

Seasonal recurrence and modular assembly of an Arctic pelagic marine microbiome

Corresponding Author: Dr Matthias Wietz

Version 0:

Reviewer comments:

Reviewer #1

(Remarks to the Author)

In this manuscript Priest T., Oldenburg E., and co-authors used in situ autonomous samplers and sensors to investigate the taxonomic and functional dynamics of a pelagic Arctic Ocean microbiome over four years from 2016 to 2020. Through long-read metagenomics combined with 16S and 18S rRNA gene sequencing, they investigated dominant prokaryotic and micro-eukaryotic populations and community gene recurrence. This revealed temporal modules representing distinct ecological states linking the prokaryotic microbiome to specific micro-eukaryotic populations and oceanographic conditions and revealed seasonal heterogeneity in environmental selection processes. This study provides very valuable datasets to study the seasonal structure and composition of difficult-to-access and sample Arctic Ocean plankton communities. It contributes to further our understanding of how these communities are selected and structured under large environmental variability in an ecosystem strongly impacted by climate change. While I think this manuscript should be considered for publication, I have some major comments/concerns regarding the analyses and the associated conclusions that I think should be addressed.

Major comments:

- Overall, I think the authors should further consider and/or discuss the role of North Atlantic water influence in shaping seasonally recurring assembly of prokaryotic and micro-eukaryotic communities. It may be interesting to tackle this question using Lagrangian analysis of satellite-derived currents.
- The population dynamics variation of prokaryotic and micro-eukaryotic communities over seasons may also be investigated at a finer resolution level than solely using the Fourier transformation-based approach. Using such an alternative approach integrating finer scale species variations in time, are similar results observed and conclusions confirmed?
- To what extent the prokaryotic microbiome is driving specific micro-eukaryotic populations, and vice versa? Temporal sampling available here can give some very interesting insights into biotic factors shaping population dynamics in the Arctic.
- More methodological details for the Fourier transform approach and correlation analyses between oscillation signals are needed. In addition, the benefit of this approach over a classical ASV-level co-occurrence analysis should be demonstrated. Where is the supplementary information describing all the details of the Fourier-transform methodology? I think linking to previous publications is not sufficient here.
- Since long read metagenomes are available there's a great opportunity for the authors to directly integrate taxonomic and functional information instead of linking them via potentially biased correlation signals. In addition, very few polar night metagenomes (< 10?) seem to be available. This disequilibrium in the number of samples covering each season could significantly bias the network modules analysis. Thus, the comparison between oscillation signals of ASV and gene function should be complemented by direct integration of taxonomic and functional information from the metagenomics data. A simple correlation metric is likely not sufficient to link taxonomy and function here as many latent factors and environmental parameters could drive these significant positive correlations. Thus, this analysis should be complemented by an independent analysis of metagenomes to directly link taxonomy and function, for example through the reconstruction of metagenome-assembled genomes, or eventually by leveraging taxonomic annotations of ORFs predicted.
- The selection pressure analysis and results may largely be driven by the possible disequilibrium in the number of metagenomes by season/module. Indeed, the authors observed that "The spring module M3 comprises a nearly two-fold larger diversity of gene functions than the polar night module M2, but fourfold less ASVs.". This reduction in gene function diversity during the polar night may be driven by the very few numbers of metagenomes available for the night periods. This potential bias should be investigated to confirm the reported results.

- In addition, the conclusion that the “selection pressure may be stronger on the function than the organismal level in spring.” may also be driven by this potential bias in the number of metagenomes for each season. Also, selection likely acts at the functional level so the authors should further argue and develop how selection can act at the species level and how this can be demonstrated.

- The disequilibrium in the number of metagenomes may also influence the functional redundancy comparison between seasons/modules, so this should be taken into account and/or corrected for in the analyses.

Minor comments:

- Legend of Figure 4 needs to be corrected as there are 4 panels in the figure and only 3 descriptions in the legend.

- Overall, although data and code are available, the detailed methodologies should accompany the manuscript.

(Remarks on code availability)

Reviewer #2

(Remarks to the Author)

This work investigates the seasonal dynamics of planktonic microbial communities in the Arctic Ocean over a four-year period. Taxonomic and functional dynamics were investigated using samples obtained via autonomous samplers coupled to in situ sensors. The authors have combined classic metabarcoding (16S and 18S) with long-read metagenomics. Results indicate recurrent dynamics of taxa as well as functional genes. In addition, five temporal prokaryotic modules were detected that may represent different configurations of the microbiota over the year. These modules had specific characteristics in terms of taxonomic and metabolic composition and were linked to specific microbial eukaryotes and environmental conditions. Overall, results contribute to a better understanding of the dynamics of polar microbiomes and the links between diversity and function.

For the most part, the manuscript is clear, although I have a number of general comments that could help to improve this work.

-One of the main advantages of doing long-read metagenomics over short reads is reconstructing high-quality metagenome-assembled genomes (MAGs), but this was not done here. These MAGs could contribute to better linking taxonomy with function and provide genomic context to the retrieved genes (ORFs). So, I suggest that the authors consider generating MAGs. Single-contig prokaryotic MAGs, including the 16S would be expected, and this could be used to link the metabarcoding 16S data with the MAGs as well as other recovered genes (ORFs).

-Generating MAGs could also contribute to retrieving eukaryotic MAGs, which could improve linking gene functions with taxa and specific dynamics that involve prokaryotes and eukaryotes.

-I also wonder why eukaryotic genes (ORFs) were not predicted, considering microbial eukaryotes are analyzed via the 18S. I suggest separating prokaryotic from eukaryotic reads (or contigs) using Tiara (<https://academic.oup.com/bioinformatics/article/38/2/344/6375939>) or EukRep (<https://github.com/patrickwest/EukRep>) and then running metaeuk (<https://github.com/soedinglab/metaeuk>) on the detected eukaryotic reads or contigs. Predicted eukaryotic genes could then be annotated (using, e.g., eggNOG mapper), and eukaryotic genes could then be included in the analyses. I think including eukaryotic genes/ORFs will strengthen the paper.

-In order to determine what proportion of the microbiota was recovered using long-reads in the investigated area, I suggest comparing the predicted genes/ORFs (the 48.5 million) to those retrieved by the TARA Arctic expedition using short-read sequencing (<https://www.sciencedirect.com/science/article/pii/S009286741931164X>). This will also contribute to validating predicted genes and, more importantly, will contribute to checking the expression of genes of interest (by, e.g., mapping unassembled metatranscriptomic reads from TARA to the metabolic genes you're interested in or just by linking your predicted genes to the gene expression information of the TARA gene catalog in the paper above). Also, this paper should be cited I think.

Minor comments:

L193: the name “gene clusters” can be confusing. “Non-redundant genes” has been used in multiple papers.

L217. See also for the Mediterranean: <https://microbiomejournal.biomedcentral.com/articles/10.1186/s40168-023-01523-z>

L220: I'd tone this down. This could happen in polar or temperate areas with strong environmental selection, but maybe (probably) not in tropical areas, where dynamics could be more stochastic.

L279: Is there any evidence that *Synechococcus* is metabolically active? TARA-Arctic RNAseq data could be used to check that (see previous comment on TARA metatranscriptomic data).

L312-6: Unclear. In particular, please clarify the link between selection and function.

L346: Check phrasing

L366: Were 2 x 500 ml seawater samples enough for PacBio sequencing? I would indicate the amounts of DNA extracted from the samples (range).

What were the time periods between sample fixation and filtering in the Sterivex? (I would indicate that in the paper, given that the methodology is different from most papers).

L383: Did the fixation with mercuric chloride have any influence on the DNA extraction?

L398: Did you consider assembling the data? (was that not done for any specific reason?). One good assembler is HiFiasm-meta (<https://www.nature.com/articles/s41592-022-01478-3>). Also, for building MAGs, this is a good pipeline: <https://github.com/PacificBiosciences/pb-metagenomics-tools/blob/master/docs/Tutorial-HiFi-MAG-Pipeline.md>

As mentioned above, I think building MAGs and predicting eukaryotic genes will strengthen this work.

Figure 2: Panels a) and b). I would mark the polar day and night periods in the figures, as in Figure 1b. I would add "Prokaryotes" and "Microeukaryotes" in the plots to facilitate visualization. I would consider plotting both prokaryotes and microeukaryotes in the same figure, to see the patterns more easily.

Figure 2C-D: It is unclear why the correlations between Richness (S), Shannon (H) and Evenness (J) are shown, as these values are related by definition ($J = H/\ln(S)$; Pielou's evenness). Maybe I am missing something here, it be good to clarify.

Figure 3. Panels a) and b). Please add "Prokaryotes" and "Microeukaryotes" in the figure to facilitate the visualization. Panel c) : It'd be very interesting to have the "Eukaryotic gene clusters" here as well.

Figure 4. Panel a) the circle and triangle nodes are barely visible; check. Also, check the panel's legends; it seems b) does not correspond to the displayed panel.

Figure 7. Include the description of each cluster as in Figure 5. Also, try to explain better this figure in the legend (it took me a while to understand it).

(Remarks on code availability)

Version 1:

Reviewer comments:

Reviewer #1

(Remarks to the Author)

I thank the authors for their diligent revision of the manuscript.

In particular, I appreciate the additional analyses to integrate taxonomy and functional information at HiFi read level, as well as the comparative analysis with "Tara Arctic" (note this should be corrected to Tara Oceans Polar Circle (TOPC) in the text) metatranscriptomics data.

For this latter analysis, the authors should discuss why the TOPC station closest to the FRAM strait (TARA_163) is actually displaying very few mapping transcripts, in particular as compared to station TARA_168, which is located in the Barents Sea (<https://www.nature.com/articles/sdata201523/figures/2>). Also, TOPC stations should be highlighted in panel a) of the new figure 6, and it should be noted that station TARA_155 is not displayed on the map.

Regarding data availability, the authors should make sure that all analyses are reproducible and should also facilitate future analyses by other teams. After some research at <https://edmond.mpg.de/dataset.xhtml?persistentId=doi:10.17617/3.CA8MQY>, it is still unclear to me how mooring data / metadata can easily be linked to

sequencing data via ENA accession numbers. If not already available, the authors should provide a file mapping the metadata in RAS_F4_META.txt to sequencing samples / ENA accession numbers.

(Remarks on code availability)

Reviewer #2

(Remarks to the Author)

All my previous comments have been addressed.

In the new Figure 6, please add the number of the corresponding TARA stations to the pie charts in Panel A (this will also help to link the information in the pie charts to the bar plots in Panel B). In Panel A, you could also indicate the mooring site shown in Fig.1.

Fig1 (map): check the inset (I see an empty white box).

(Remarks on code availability)

Responses to reviewer's comments

Reviewer 1

In this manuscript Priest, Oldenburg and co-authors used in situ autonomous samplers and sensors to investigate the taxonomic and functional dynamics of a pelagic Arctic Ocean microbiome over four years from 2016 to 2020. Through long-read metagenomics combined with 16S and 18S rRNA gene sequencing, they investigated dominant prokaryotic and micro-eukaryotic populations and community gene recurrence. This revealed temporal modules representing distinct ecological states linking the prokaryotic microbiome to specific micro-eukaryotic populations and oceanographic conditions and revealed seasonal heterogeneity in environmental selection processes. This study provides very valuable datasets to study the seasonal structure and composition of difficult-to-access and sample Arctic Ocean plankton communities. It contributes to further our understanding of how these communities are selected and structured under large environmental variability in an ecosystem strongly impacted by climate change. While I think this manuscript should be considered for publication, I have some major comments/concerns regarding the analyses and the associated conclusions that I think should be addressed.

We thank the reviewer for the positive assessment and constructive comments. Please find our detailed answers below.

- Overall, I think the authors should further consider and/or discuss the role of North Atlantic water influence in shaping seasonally recurring assembly of prokaryotic and micro-eukaryotic communities. It may be interesting to tackle this question using Lagrangian analysis of satellite-derived currents.

Thank you for the suggestion. Indeed, previous research on the physical-oceanographic features of the West Spitsbergen Current has revealed changes in current velocity and magnitude over time. However, the observed changes are either a) decadal shifts, or b) interannual variability. To our knowledge, there has not been any evidence for recurrent seasonal shifts in the physical properties of the WSC. For this reason, we consider that this is likely to have a minor impact on shaping the seasonal structuring of the microbiome.

- The population dynamics variation of prokaryotic and micro-eukaryotic communities over seasons may also be investigated at a finer resolution level than solely using the Fourier transformation-based approach. Using such an alternative approach integrating finer scale species variations in time, are similar results observed and conclusions confirmed?

We agree that there is a great interest to go beyond Fourier-transformed abundances and study the fine-scale resolution of population oscillations. Nonetheless, we feel that this goes beyond the scope of this study, which was to elucidate seasonal and inter-annual trends. The choice to employ Fourier Transformation was to enable the elucidation of oscillations in population dynamics over seasonal, multi-year scales with infrequent sampling resolution. Research on finer scale dynamics requires consistent daily to weekly sampling, which was impossible with our mooring instrumentation.

- To what extent the prokaryotic microbiome is driving specific micro-eukaryotic populations, and vice versa? Temporal sampling available here can give some very interesting insights into biotic factors shaping population dynamics in the Arctic.

Elucidating the extent of influence is a particularly challenging task; at best, only able through cross-convergence networks. Although this would be a strategy we could employ, we have

now instead integrated microeukaryotic ASVs into the network analysis, in light of several other comments. With this, we gain more information about the connection between trophic levels and overall genetic repertoire, without relying on post-network abundance correlations. Although this does not directly address influence, it provides a more holistic perspective on microbiome dynamics, considering all taxonomic and functional information obtainable from metagenomics and metabarcoding.

- More methodological details for the Fourier transform approach and correlation analyses between oscillation signals are needed. In addition, the benefit of this approach over a classical ASV-level co-occurrence analysis should be demonstrated. Where is the supplementary information describing all the details of the Fourier-transform methodology? I think linking to previous publications is not sufficient here.

We have added a more detailed description of the Fourier Transformation approach to the supplementary information.

- Since long read metagenomes are available there's a great opportunity for the authors to directly integrate taxonomic and functional information instead of linking them via potentially biased correlation signals. In addition, very few polar night metagenomes (< 10?) seem to be available. This disequilibrium in the number of samples covering each season could significantly bias the network modules analysis. Thus, the comparison between oscillation signals of ASV and gene function should be complemented by direct integration of taxonomic and functional information from the metagenomics data. A simple correlation metric is likely not sufficient to link taxonomy and function here as many latent factors and environmental parameters could drive these significant positive correlations. Thus, this analysis should be complemented by an independent analysis of metagenomes to directly link taxonomy and function, for example through the reconstruction of metagenome-assembled genomes, or eventually by leveraging taxonomic annotations of ORFs predicted.

We agree that correlation-based approaches are insufficient for inferring coupled or decoupled dynamics in taxonomy and function. In the revised manuscript, we have included information on the taxonomic affiliation of genes via BLAST and last-common ancestor analysis on HiFi read level, and orphaning the classifications to the contained genes. We believe that this approach is far more thorough than relying on metagenome-assembled genomes to link taxonomy and function, as metagenome-assembled genomes often only capture a fraction of the sequenced genetic material. We have included a new paragraph (lines 222–237) with further details. In addition, we have included a mention of the gene taxonomies, where relevant, when reporting module functions, e.g. lines 268–270, 285–288, 298–300. In addition, we have added a table showing the connection between gene clusters and taxonomy in the paper's data repository (<https://doi.org/10.17617/3.CA8MQY>).

- The selection pressure analysis and results may largely be driven by the possible disequilibrium in the number of metagenomes by season/module. Indeed, the authors observed that “The spring module M3 comprises a nearly two-fold larger diversity of gene functions than the polar night module M2, but fourfold less ASVs”. This reduction in gene function diversity during the polar night may be driven by the very few numbers of metagenomes available for the night periods. This potential bias should be investigated to confirm the reported results.

Although we are unable to account for the possible changes that may occur if more material from polar night samples could have been sequenced, we can assess if sequencing depth and sampling frequency impacts our findings. To achieve this, we employed a combination of

subsampling (rarefying) and subsetting the dataset to determine if this had an influence on the module dynamics. For instance, we subsetted the functional and taxonomic profiles to include only one sample per calendar month, and subsampled to the lowest count observed in each profile – demonstrating only a negligible change in the diversity and composition of module components. This finding also confirms that the patterns we are capturing reflect the major taxonomic and functional shifts, while those in the rare biosphere or more ephemeral shifts are likely undetectable with our methodology and sampling resolution. To summarise this, we have included additional text (Lines 256–261) as well as Supplementary Figure S2.

- In addition, the conclusion that the “selection pressure may be stronger on the function than the organismal level in spring.” may also be driven by this potential bias in the number of metagenomes for each season. Also, selection likely acts at the functional level so the authors should further argue and develop how selection can act at the species level and how this can be demonstrated.

Please see our response above regarding the bias in metagenome numbers. In addition, our aim of the final section of the manuscript was to provide insights into how selection may change over seasons. As you say in your comment with the phrase “selection **likely** acts...”, this is a topic that few environmental studies have addressed. Although we cannot be conclusive on the extent and target of selection pressure, our findings provide evidence-based propositions as to how selection pressure may be heterogeneous over time. We feel that demonstrating species-level selection would require far more data (e.g. obtainable via higher sampling frequency or metatranscriptomics), which was impossible with our mooring instrumentation.

- The disequilibrium in the number of metagenomes may also influence the functional redundancy comparison between seasons/modules, so this should be taken into account and/or corrected for in the analyses.

Please see response to above comments that address the impact of unequal temporal distribution of metagenomic samples.

- Legend of Figure 4 needs to be corrected as there are 4 panels in the figure and only 3 descriptions in the legend.

Changed accordingly.

- Overall, although data and code are available, the detailed methodologies should accompany the manuscript.

Additional information on methodologies has been included in a new file (Supplementary Information).

Reviewer #2

This work investigates the seasonal dynamics of planktonic microbial communities in the Arctic Ocean over a four-year period. Taxonomic and functional dynamics were investigated using samples obtained via autonomous samplers coupled to in situ sensors. The authors have combined classic metabarcoding (16S and 18S) with long-read metagenomics. Results indicate recurrent dynamics of taxa as well as functional genes. In addition, five temporal prokaryotic modules were detected that may represent different configurations of the microbiota over the year. These modules had specific characteristics in terms of taxonomic

and metabolic composition and were linked to specific microbial eukaryotes and environmental conditions. Overall, results contribute to a better understanding of the dynamics of polar microbiomes and the links between diversity and function. For the most part, the manuscript is clear, although I have a number of general comments that could help to improve this work.

We thank the reviewer for the positive assessment and constructive comments. Please find our detailed answers below.

- One of the main advantages of doing long-read metagenomics over short reads is reconstructing high-quality metagenome-assembled genomes (MAGs), but this was not done here. These MAGs could contribute to better linking taxonomy with function and provide genomic context to the retrieved genes (ORFs). So, I suggest that the authors consider generating MAGs. Single-contig prokaryotic MAGs, including the 16S would be expected, and this could be used to link the metabarcoding 16S data with the MAGs as well as other recovered genes (ORFs).

Thank you for the suggestion. Although we agree, and have previously published research, showcasing the relevance of metagenome-assembled genomes based on long-read technologies, the primary aim of this study was to move beyond a MAG-centric approach. Although MAGs are powerful for connecting taxonomy to function and integrating environmental and evolutionary dynamics, they often only represent a fraction of the sequenced material. Therefore, relying solely on MAGs can limit our perspectives on the dynamics in microbiome taxonomy and function. In this study, we aimed to showcase that long-read metagenomes also enable a complete assessment of microbiome taxonomy and function, independent of recovering MAGs. To support this, and to address your comment, we have generated MAGs from the metagenomes and included a new paragraph justifying our approach (lines 222–237), and included the new Supplementary Figure S1. In addition, we have leveraged the length and quality of PacBio HiFi reads for taxonomic classification on the reads themselves, using a BLAST and last common ancestor approach. Read taxonomies were subsequently orphaned to the contained genes, allowing us to connect taxonomy and function directly on the gene level. We have included a mention of the gene taxonomies, where relevant, when reporting module functions (e.g. lines 268–270, 285–288, 298–300).

- Generating MAGs could also contribute to retrieving eukaryotic MAGs, which could improve linking gene functions with taxa and specific dynamics that involve prokaryotes and eukaryotes.

Please see the above response. In addition, sequencing depth was unfortunately too low for recovering eukaryotic MAGs. This is evident when considering the domain-level composition of the metagenomic reads, with only 0.27 out of 8.7 million reads identified as eukaryotic.

- I also wonder why eukaryotic genes (ORFs) were not predicted, considering microbial eukaryotes are analyzed via the 18S. I suggest separating prokaryotic from eukaryotic reads (or contigs) using Tiara (<https://academic.oup.com/bioinformatics/article/38/2/344/6375939>) or EukRep (<https://github.com/patrickwest/EukRep>) and then running metaeuk (<https://github.com/soedinglab/metaeuk>) on the detected eukaryotic reads or contigs. Predicted eukaryotic genes could then be annotated (using, e.g., eggnoG mapper), and eukaryotic genes could then be included in the analyses. I think including eukaryotic genes/ORFs will strengthen the paper.

Thank you for the suggestion. In the first manuscript version, we had employed Tiara to subset the reads to those of prokaryotic origin before the gene and function analysis. Given your

comment, and several other comments we received, we have now incorporated the entire gene pool predicted from the metagenomes in our analysis. Based on read-level domain classification, only 0.27 out of 8.7 million reads were associated with Eukaryotes. This resulted in tens of thousands of eukaryotic genes being incorporated into the analysis. However, unfortunately, few eukaryotic genes ended up in our network modules, which we believe is due to insufficient sequencing depth.

- In order to determine what proportion of the microbiota was recovered using long-reads in the investigated area, I suggest comparing the predicted genes/ORFs (the 48.5 million) to those retrieved by the TARA Arctic expedition using short-read sequencing (<https://www.sciencedirect.com/science/article/pii/S009286741931164X>). This will also contribute to validating predicted genes and, more importantly, will contribute to checking the expression of genes of interest (by, e.g., mapping unassembled metatranscriptomic reads from TARA to the metabolic genes you're interested in or just by linking your predicted genes to the gene expression information of the TARA gene catalog in the paper above). Also, this paper should be cited I think.

We agree that Tara Oceans data is highly valuable for comparative analysis. In the revised manuscript, we hence leveraged the metatranscriptomic data from Tara Arctic to assess the transcription of module functions across the wider Arctic Ocean. We have included a new section in the results and discussion on our findings from this analysis (Lines 315–337) as well as a new Figure 6.

L193: the name “gene clusters” can be confusing. “Non-redundant genes” has been used in multiple papers.

We feel that “gene clusters” is more accurate terminology. The term “non-redundant genes” is typically used when referring to identical or near-identical genes (>99%) and thus may lead to more ambiguity.

L217. See also for the Mediterranean: <https://microbiomejournal.biomedcentral.com/articles/10.1186/s40168-023-01523-z>

Thank you, we have added this reference.

L220: I'd tone this down. This could happen in polar or temperate areas with strong environmental selection, but maybe (probably) not in tropical areas, where dynamics could be more stochastic.

Sentence has been modified, see lines 215–218.

L279: Is there any evidence that *Synechococcus* is metabolically active? TARA-Arctic RNAseq data could be used to check that (see previous comment on TARA metatranscriptomic data).

After incorporating eukaryotic information into the network analysis, the *Synechococcus* signal is no longer pronounced. For this reason, we have removed the related sentences.

L312-6: Unclear. In particular, please clarify the link between selection and function.

We have reworded this paragraph to improve clarity.

L346: Check phrasing

Thank you, phrasing has been modified.

L366: Were 2 x 500 ml seawater samples enough for PacBio sequencing? I would indicate the amounts of DNA extracted from the samples (range).

Across all 97 samples, we obtained 0.003–10 ng DNA per μL elution buffer (average 1.4 ng). Scaled to the total elution volume of 60 μL , this corresponded to a total yield of 0.2–590 ng DNA per sample. These were always sufficient for amplicon analyses, supported by rarefaction curves showing fully covered diversity. Samples selected for PacBio sequencing had 0.01–10 ng DNA per μL , and underwent strict quality control. This information has been added to lines 425–427.

What were the time periods between sample fixation and filtering in the Sterivex? (I would indicate that in the paper, given that the methodology is different from most papers).

The period varied depending on the time of sampling during the annual cycle. Samples collected soon after mooring deployment remained up to 12 months in the sampling bags at in situ temperature. However (also see following comment), the effects of fixation on community composition are immediate and do not escalate further over time – hence, the period between sample collection/fixation and filtration do not matter. We have added this information to lines 406–408.

L383: Did the fixation with mercuric chloride have any influence on the DNA extraction?

Expanding on the previous comment: in a prior study ([doi:10.3389/fmicb.2022.999925](https://doi.org/10.3389/fmicb.2022.999925); now cited), we have assessed the influence of fixation on DNA extraction and sequencing in time-series studies. Fixation does not change the overarching patterns, despite some influence on the rare biosphere. We are hence confident in the validity of our results, supported by their strong connection with environmental conditions. We have added this information to lines 406–408.

L398: Did you consider assembling the data? (was that not done for any specific reason?). One good assembler is HiFiasm-meta (<https://www.nature.com/articles/s41592-022-01478-3>). Also, for building MAGs, this is a good pipeline: <https://github.com/PacificBiosciences/pb-metagenomics-tools/blob/master/docs/Tutorial-HiFi-MAG-Pipeline.md> As mentioned above, I think building MAGs and predicting eukaryotic genes will strengthen this work.

Please see response to your earlier comment about the MAG generation.

Figure 2: Panels a) and b). I would mark the day and night in the figures, as in Figure 1b. I would add “Prokaryotes” and “Microeukaryotes” in the plots to facilitate visualization. I would consider plotting both prokaryotes and microeukaryotes in the same figure, to see the patterns more easily. Figure 2C-D: It is unclear why the correlations between Richness (S), Shannon (H) and Evenness (J) are shown, as these values are related by definition ($J = H/\ln(S)$; Pielou’s evenness). Maybe I am missing something here, it be good to clarify.

We apologise if this was unclear. Of course, the three metrics are intrinsically related, as Shannon diversity is calculated from Richness and Evenness. However, often studies only discuss changes in Shannon Diversity, which could be driven by changes in Richness or Evenness or both. By assessing all three metrics, it allows for deeper insights into diversity

dynamics; i.e. whether Shannon diversity changes only due to a shift in the evenness or due to an increased richness.

Figure 3. Panels a) and b). Please add “Prokaryotes” and “Microeukaryotes” in the figure to facilitate the visualization. Panel c): It’d be very interesting to have the “Eukaryotic gene clusters” here as well.

We have included Prokaryotes and Microeukaryotes as labels into panels a) and b). In the revised manuscript, we have now incorporated eukaryotic gene clusters. For panel c), the gene clusters now represent the combination of both prokaryotic and microeukaryotic genes.

Figure 4. Panel a) the circle and triangle nodes are barely visible; check. Also, check the panel’s legends; it seems b) does not correspond to the displayed panel.

The figure legend has been corrected. Increasing the discernibility of triangles and circles in the network is, unfortunately, very challenging, given the >8000 points included.

Figure 7. Include the description of each cluster as in Figure 5. Also, try to explain better this figure in the legend (it took me a while to understand it).

A legend has now been included for the modules. The figure legend has been revised to improve clarity.

Reviewer #1

I thank the authors for their diligent revision of the manuscript. In particular, I appreciate the additional analyses to integrate taxonomy and functional information at HiFi read level, as well as the comparative analysis with “Tara Arctic” (note this should be corrected to Tara Oceans Polar Circle (TOPC) in the text) metatranscriptomics data. For this latter analysis, the authors should discuss why the TOPC station closest to the FRAM strait (TARA_163) is actually displaying very few mapping transcripts, in particular as compared to station TARA_168, which is located in the Barents Sea (<https://www.nature.com/articles/sdata201523/figures/2>).

We have now included an explanation as to why the Tara Station in the Fram Strait has reduced transcription detected compared to the Barents Sea station (lines 336 - 341).

Also, TOPC stations should be highlighted in panel a) of the new figure 6, and it should be noted that station TARA_155 is not displayed on the map.

The map has been recreated to include station Tara_155 and all stations have now been labelled in the map.

Regarding data availability, the authors should make sure that all analyses are reproducible and should also facilitate future analyses by other teams. After some research at <https://edmond.mpg.de/dataset.xhtml?persistentId=doi:10.17617/3.CA8MQY>, it is still unclear to me how mooring data / metadata can easily be linked to sequencing data via ENA accession numbers. If not already available, the authors should provide a file mapping the metadata in RAS_F4_META.txt to sequencing samples / ENA accession numbers.

Thank you for pointing this out. We have now included an additional supplementary table (no. 2) that maps the sample names, and thus the metadata, to the ENA accession numbers. This table has also been included on the data repository.

Reviewer #2

All my previous comments have been addressed.

In the new Figure 6, please add the number of the corresponding TARA stations to the pie charts in Panel A (this will also help to link the information in the pie charts to the bar plots in Panel B). In Panel A, you could also indicate the mooring site shown in Fig.1.

Station labels have now been included in the map along with a star to indicate the location of the mooring station.

Fig1 (map): check the inset (I see an empty white box).

Thank you for pointing this out. We are unsure as to why you can not see the inset but we have re-exported the figure and checked that the content is visible.